# DynDepNet: Learning Time-Varying Dependency Structures from fMRI Data via Dynamic Graph Structure Learning

**Alexander Campbell** [1 2]   **Antonio Giuliano Zippo** [3]   **Luca Passamonti** [1]   **Nicola Toschi** [4 5]   **Pietro Liò** [1]

## Abstract

Graph neural networks (GNNs) have demonstrated success in learning representations of brain graphs derived from functional magnetic resonance imaging (fMRI) data. However, existing GNN methods assume brain graphs are static over time and the graph adjacency matrix is known prior to model training. These assumptions contradict evidence that brain graphs are time-varying with a connectivity structure that depends on the choice of functional connectivity measure. Incorrectly representing fMRI data with noisy brain graphs can adversely affect GNN performance. To address this, we propose DynDepNet, a novel method for learning the optimal time-varying dependency structure of fMRI data induced by downstream prediction tasks. Experiments on real-world fMRI datasets, for the task of sex classification, demonstrate that DynDepNet achieves state-of-the-art results, outperforming the best baseline in terms of accuracy by approximately 8 and 6 percentage points, respectively. Furthermore, analysis of the learned dynamic graphs reveals prediction-related brain regions consistent with existing neuroscience literature.

## 1. Introduction

Functional magnetic resonance imaging (fMRI) is primarily used to measure blood-oxygen level dependent (BOLD) signal (or blood flow) in the brain (Huettel et al., 2004). It is one of the most frequently used non-invasive imaging techniques for investigating brain function (Power et al., 2014; Just & Varma, 2007). Typically, this is achieved by employing a statistical measure of pairwise dependence, such as Pearson correlation or mutual information. These measures are used to summarize the functional connectivity (FC) between BOLD signals originating from anatomically separate brain regions (Friston, 1994). The resulting FC matrices, or functional connectomes, have found extensive application in graph-based network analyses (Sporns, 2022; Wang et al., 2010) and as inputs to machine learning models (He et al., 2020) offering valuable insights into both normal and abnormal brain function.

Graph neural networks (GNNs) are a type of deep neural network that can learn representations of graph-structured data (Wu et al., 2020a). GNNs employ a message-passing scheme to learn these representations by aggregating information from the neighborhoods of nodes, utilizing the observed graph structure or adjacency matrix. By preprocessing FC matrices into adjacency matrices that correspond to brain graphs, GNNs have recently exhibited success in various fMRI-related prediction tasks. These tasks include predicting phenotypes such as biological sex (Azevedo et al., 2022; Kim & Ye, 2020) and age (Gadgil et al., 2020), analyzing brain activity during different cognitive tasks (Zhang et al., 2021), and identifying brain disorders like autism spectrum disorder (Li et al., 2021) and attention deficit hyperactivity disorder. (Zhao et al., 2022).

However, the majority of existing GNN methods applied to fMRI data rely on two key assumptions: (1) brain graphs are static and not time-varying, and (2) the true dependency structure between brain regions is known prior to model training. Although these assumptions are convenient for implementation purposes, they contradict a growing body of evidence suggesting FC dynamically changes over time (Calhoun et al., 2014; Chang & Glover, 2010), and that no one statistical measure of dependency exists for truly capturing FC (Mohanty et al., 2020). In order to ensure GNNs can effectively learn meaningful representations for use in downstream tasks, it is crucial to establish an approach for constructing the dependency structure of dynamic graphs that accurately reflects the underlying fMRI data.

[1]Department of Computer Science and Technology, University of Cambridge, Cambridge, United Kingdom [2]The Alan Turing Institute, London, United Kingdom [3]Institute of Neuroscience, University of Milano-Bicocca, Milan, Italy [4]University of Rome Tor Vergata, Rome, Italy [5]A.A. Martinos Center for Biomedical Imaging, Harvard Medical School, Boston, United States. Correspondence to: Alexander Campbell <ajrc4@cl.cam.ac.uk>.

*Workshop on Interpretable ML in Healthcare at International Conference on Machine Learning (ICML)*, Honolulu, Hawaii, USA. 2023. Copyright 2023 by the author(s).

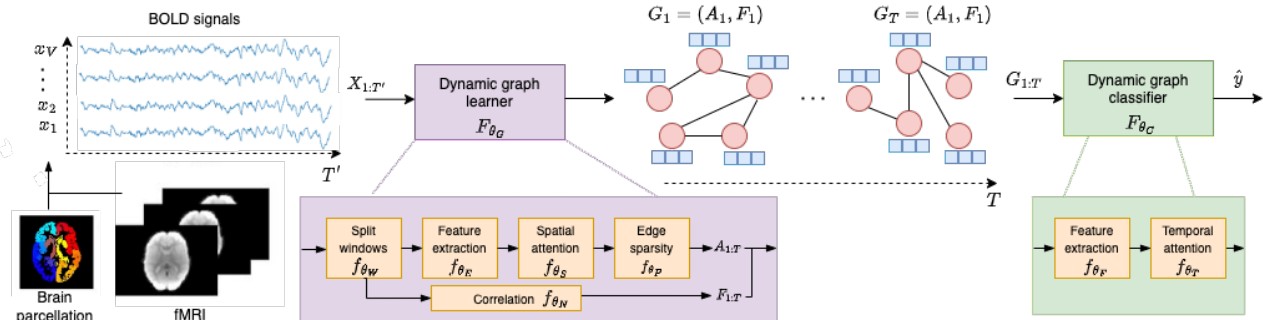

*Figure 1.* The conceptual framework of DynDepNet $F_\theta(\cdot) = F_{\theta_C} \circ F_{\theta_G}(\cdot)$ which consists of a dynamic graph learner $F_{\theta_G}(\cdot)$ and a dynamic graph classifier $F_{\theta_C}(\cdot)$. The dynamic graph learner takes BOLD signals from fMRI data as input $\mathbf{X}_{1:T'} \in \mathbb{R}^{V \times T'}$ and outputs a dynamic brain graph $G_{1:T} = (\mathbf{A}_{1:T}, \mathbf{F}_{1:T}) = F_{\theta_G}(\mathbf{X}_{1:T'})$ where $\mathbf{A}_{1:T} \in (0,1)^{V \times V \times T}$ is a dynamic adjacency matrix, $\mathbf{F}_{1:T} \in \mathbb{R}^{V \times B \times T}$ is dynamic node feature matrix, and $T \leq T'$. The dynamic graph classifier predicts class labels using the learnt dynamic brain graph $\hat{y} = F_{\theta_C}(G_{1:T})$. Each of these two components are further spit into individual modules describing their purpose.

**Contributions** We propose DynDepNet[1], the first end-to-end trainable GNN-based model designed to learn task-specific dynamic brain graphs from fMRI data in a supervised manner. DynDepNet addresses the limitations of existing methods by constructing dynamic graph adjacency matrices using spatially attended brain region embeddings derived from windowed BOLD signals. Furthermore, DynDepNet incorporates temporal attention and learnable edge sparsity to enhance classification performance and interpretability. Through extensive experiments on real-world resting-state and task fMRI datasets, DynDepNet achieves state-of-the-art performance in the task of biological sex classification. Additionally, an interpretability analysis of the learned dynamic graph adjacency matrices reveals prediction-related brain regions that align with existing neuroscience literature.

## 2. Related work

**Brain graph classification** Recent methods for brain graph classification employ static measures of FC such as Pearson correlation (Zhang et al., 2021; Kim & Ye, 2020; Azevedo et al., 2022; Huang et al., 2022) or partial correlation (Li et al., 2021) for graph construction. However, this approach overlooks the dynamic nature of FC, which is known to change over time (Calhoun et al., 2014), and the fact that the choice of FC measure affects the performance on downstream tasks (Korhonen et al., 2021). To capture the non-stationary nature of FC, recent methods have emerged that specifically aim to learn representations of dynamic brain graphs. For instance, Gadgil et al. (2020) propose a variant of the spatial-temporal GNN (Yan et al., 2018) for fMRI data. However, only graph node features are

[1]Code available at https://github.com/ajrcampbell/DynDepNet

taken as time-varying whilst the adjacency matrix remains static. Similarly, Kim et al. (2021) leverage spatial-temporal attention within a transformer framework (Vaswani et al., 2017) to classify dynamic brain graphs. However, the graph adjacency matrix is assumed to be unweighted, and similar to previous methods, still requires careful selection of a FC measure prior to model training.

**Graph structure learning** Deep graph structure learning (GSL) (Zhu et al., 2021; Kalofolias et al., 2017) is a method for addressing the challenge of defining the correct prior dependency structure for GNNs. GSL methods learn the adjacency matrix of a dataset alongside the parameters of a GNN by solving a downstream task like classification or regression. While recent GSL attempts on multivariate time-series data have shown improved performance in regression tasks (Cao et al., 2020; Wu et al., 2020b; 2019), they typically learn a single static graph for all samples, which is not suitable for multi-subject fMRI data where subject-level brain graphs exhibit discriminative differences (Finn et al., 2015). Several fMRI-specific GSL methods, such as those proposed by Mahmood et al. (2021), Riaz et al. (2020), and Kan et al. (2022), learn brain graphs from BOLD signals for downstream classification tasks. However, unlike DynDepNet, these methods assume static brain graphs.

## 3. Problem formulation

We formulate the problem of dynamic brain graph structure learning as a supervised multivariate timeseries classification task. Let $\mathbf{X}_{1:T'} = (\mathbf{x}_1, \ldots, \mathbf{x}_{T'}) \in \mathbb{R}^{V \times T'}$ denote BOLD signals from $V \in \mathbb{N}$ brain regions measured over $T' \in \mathbb{N}$ timepoints, and let $y \in [0, \ldots C-1]$ represent a corresponding discrete class label, where $C \in \mathbb{N}$ is the total number of classes. We assume that $\mathbf{X}_{1:T'}$ posses a

true, yet unknown, nonstationary dependency structure. To summarize this dynamic dependency structure, we define a dynamic brain graph $G_{1:T} = (\mathbf{A}_{1:T}, \mathbf{F}_{1:T})$, which consists of a dynamic adjacency matrix $\mathbf{A}_{1:T} \in (0, 1)^{V \times V \times T}$ and a dynamic node feature matrix $\mathbf{F}_{1:T} \in \mathbb{R}^{V \times B \times T}$ over $T \leq T'$ snapshots, where $B \in \mathbb{N}$ denotes the number of features.

Given a dataset $\mathcal{D} \subset \mathcal{X} \times \mathcal{Y}$ consisting of $N \in \mathbb{N}$ subjects data $(\mathbf{X}_{1:T'}, y) \in \mathcal{D}$, we aim to train a model $F_\theta(\cdot) = F_{\theta_G} \circ F_{\theta_C}(\cdot)$ with parameters $\theta = \theta_G \cup \theta_C$ that can predict class labels $\hat{y}$ given input $\mathbf{X}_{1:T'}$ using an intermediary learnt dynamic brain graph $F_\theta(\mathbf{X}_{1:T'}) = F_{\theta_C}(F_{\theta_G}(\mathbf{X}_{1:T'})) = F_{\theta_C}(G_{1:T}) = \hat{y}$. Training consists of minimizing the discrepancy between the actual label $y$ and the predicted label $\hat{y}$, described by a loss function $\mathcal{L}(y, \{\hat{y}, G_{1:T}\})$. The optimization objective is therefore $\theta^* = \arg\min_\theta \mathbb{E}_{(\mathbf{X}_{1:T'}, y) \in \mathcal{D}}[\mathcal{L}(y, F_\theta(\mathbf{X}_{1:T'}))]$.

## 4. Method

We present DynDepNet, a novel method for learning the optimal time-varying dependency structure of fMRI data induced by a downstream prediction task. DynDepNet consists of two main components: (1) dynamic graph learner $F_{\theta_G} : \mathcal{X} \to \mathcal{G}$, and (2) dynamic graph classifier $F_{\theta_C} : \mathcal{G} \to \mathcal{Y}$. As depicted in Figure 1, we further split these two components into individual modules, elucidating their purpose, architecture, and role in learning the optimal time-varying dependency structure of fMRI data.

### 4.1. Dynamic graph learner

**Split windows** The dynamic graph learner maps BOLD signals onto a dynamic brain graph, denoted $F_{\theta_G}(\mathbf{X}_{1:T'}) = G_{1:T}$. To achieve this, the input $\mathbf{X}_{1:T'}$ is first split into windows using a temporal-splitting stack transformation $f_{\theta_W}(\cdot)$, as defined by

$$f_{\theta_W}(\mathbf{X}_{1:T'}) = \tilde{\mathbf{X}}_{1:T} = (\tilde{\mathbf{X}}_1, \dots \tilde{\mathbf{X}}_T) \in \mathbb{R}^{P \times V \times T}$$
$$\tilde{\mathbf{X}}_t = \mathbf{X}_{tS:tS+P}, \quad t = 1, \dots, T \tag{1}$$

where $1 \leq P, S \leq T'$ are hyperparameters specifying window length and stride, respectively, $\tilde{\mathbf{X}}_t \in \mathbb{R}^{P \times V}$ is a window of BOLD signal, and $T = \lfloor (T' - 2(P-1) - 1)/S + 1 \rfloor$ is the number of windows. The hyperparameters $P$ and $S$ are chosen such that each window has a stationary dependency structure. We leave more data-driven methods for selecting optimal values of $P$ and $S$, such as statistical tests for stationarity (Dickey & Fuller, 1981), for future work.

**Temporal feature extraction** The windowed BOLD signals $\tilde{\mathbf{X}}_{1:T}$ from (1) are then passed through a 2D convolutional neural network (CNN) denoted $f_{\theta_E}(\cdot)$ to extract $K_E$-dimensional feature embeddings for each brain region

independently such that $f_{\theta_E}(\tilde{\mathbf{X}}_{1:T}) = \mathbf{H}_{1:T}^G \in \mathbb{R}^{K_E \times V \times T}$. To accomplish this, we employ an inception temporal convolutional network (I-TCN) architecture inspired by Wu et al. (2020b). Our version of I-TCN incorporates dilated convolutional kernels and causal padding from original temporal convolutional network (Bai et al., 2018) and combines it with a multi-channel feature extraction in a inception structure (Szegedy et al., 2015). Formally, if $f_{\theta_E}(\cdot)$ consists of $L_G$ layers, for the $l$-th layer with $M$ convolutional filters, we have

$$\mathbf{H}_{1:T}^{(l)} = \text{ReLU}\Big(\text{BatchNorm}\big(\mathbf{H}_{1:T}^{(l-1)}$$
$$+ \|_{m=1}^M \mathbf{H}_{1:T}^{(l-1)} *_d \mathbf{W}_m^{(l)}\big)\Big) \tag{2}$$

where $\mathbf{H}_{1:T}^{(l-1)}, \mathbf{H}_{1:T}^{(l)} \in \mathbb{R}^{K_E \times V \times T}$ represent input and output embeddings, respectively. Additionally, $\mathbf{W}_m^{(l)} \in \mathbb{R}^{\lfloor K_E/M \rfloor \times K_E \times 1 \times S_m}$ denotes the $m$-th 2D convolutional filter of length $S_m$, and $\mathbf{H}_{1:T}^{(0)} = \tilde{\mathbf{X}}_{1:T}, \mathbf{H}_{1:T}^{(L_G)} = \mathbf{H}_{1:T}^G$. The symbols $\|$ and $*_d$ denote concatenation along the feature dimension and convolution operator with dilation factor $d > 0$, respectively. The functions BatchNorm$(\cdot)$ and ReLU$(\cdot)$ denote batch normalization (Ioffe & Szegedy, 2015) and a rectified linear unit activation (Nair & Hinton, 2010), respectively. To increase the receptive field size of each convolutional kernel exponentially with the number of layers, we set $d = 2^{l-1}$ following the approach of Oord et al. (2016). Moreover, we enforce $S_1 < \cdots < S_M$ to allow for the simultaneous use of small and large kernel lengths within a single layer, thereby capturing short and long-term temporal patterns efficiently during training.

**Dynamic adjacency matrix** To capture spatial relationships between brain regions, while considering the independently learned embeddings $\mathbf{H}_{1:T}^G$ from the feature extractor (2), we employ a self-attention mechanism denoted as $f_{\theta_S}(\cdot)$. Specifically, at each snapshot we utilize the embeddings to learn the pair-wise dependency structure between brain regions, denoted $\mathbf{A}_t$, using a simplified version of scaled dot-product self-attention (Vaswani et al., 2017). More formally,

$$\mathbf{A}_t = f_{\theta_S}(\mathbf{H}_t^G) = \text{Sigmoid}\left(\frac{\mathbf{Q}_t \mathbf{K}_t^\top}{\sqrt{K_S}}\right) \tag{3}$$
$$\mathbf{Q}_t = \mathbf{H}_t^G \mathbf{W}_Q, \quad \mathbf{K}_t = \mathbf{H}_t^G \mathbf{W}_K$$

where $\mathbf{Q}_t, \mathbf{K}_t \in \mathbb{R}^{V \times K_S}$ represent query and key matrices, respectively. They are obtained by performing $K_S$-dimensional linear projections on the embedding $\mathbf{H}_t^G$ using trainable matrices $\mathbf{W}_Q, \mathbf{W}_K \in \mathbb{R}^{K_E \times K_S}$. We interpret each $\mathbf{A}_t$ as a brain graph adjacency matrix, which by definition is a weighted and directed matrix that represents the connectivity between brain regions. To conform with the commonly assumed undirected nature of

brain graph analysis (Friston, 1994), we can simply set $\mathbf{W}_Q = \mathbf{W}_K$ to make $\mathbf{A}_t$ undirected. By computing self-attention matrices over the entire sequence of feature embeddings, we obtain a dynamic adjacency matrix $\mathbf{A}_{1:T} = (\mathbf{A}_1, \ldots, \mathbf{A}_T) \in (0,1)^{V \times V \times T}$ that summarizes the dynamic FC between brain regions.

**Edge sparsity** By definition, the adjacency matrix $\mathbf{A}_t$ from (3) represents a fully-connected graph at every snapshot. However, fully-connected graphs are challenging to interpret and computationally expensive for learning downstream tasks with GNNs. Moreover, they are susceptible to noise. To address these issues, we propose a soft threshold operator (Donoho, 1995) denoted as $f_{\theta_P}(\cdot)$ to enforce edge sparsity. This operator is defined following

$$f_{\theta_P}(a_{i,j,t}) = \text{ReLU}(a_{i,j,t} - \text{Sigmoid}(\theta_P)) \quad (4)$$

where $\text{Sigmoid}(\theta_P) \in (0,1)$ represents a learnable edge weight threshold, and $a_{i,j,t} \in \mathbf{A}_{1:T}$. Notably, when $a_{i,j,t} \leq \text{Sigmoid}(\theta_p)$, the output $f_{\theta_P}(a_{i,j,t})$ becomes zero, resulting in edge sparsity. To ensure the threshold $\text{Sigmoid}(\theta_P)$ starts close to 0, we initialize $\theta_P \approx -10$. This initialization prevents excessive sparsification of $\mathbf{A}_t$ in the early stages of training. Unlike other methods that rely on absolute thresholds (Zhao et al., 2021), percentile thresholds (Kim et al., 2021; Li et al., 2019), or $k$-nearest-neighbor ($k$-NN) approaches (Yu et al., 2020; Wu et al., 2020b), which require careful hyperparameter tuning to determine the optimal level of edge sparsity, our soft threshold layer learns an optimal threshold smoothly during training that is task specific.

**Dynamic node feature matrix** We take the windowed timeseries $\tilde{\mathbf{X}}_t$ from (1) and compute node feature $f_{\theta_N}(\tilde{\mathbf{X}}_t) = \mathbf{F}_t \in \mathbb{R}^{V \times B}$ as follows

$$\begin{aligned} \boldsymbol{\Sigma}_t &= \frac{1}{P-1} \tilde{\mathbf{X}}_t^\top (\mathbf{I}_P - \frac{1}{P} \mathbf{1}_P^\top \mathbf{1}_P) \tilde{\mathbf{X}}_t \\ \tilde{\mathbf{D}}_t &= \sqrt{\text{diag}(\boldsymbol{\Sigma}_t)}, \quad \mathbf{F}_t = \tilde{\mathbf{D}}_t^{-1} \boldsymbol{\Sigma}_t \tilde{\mathbf{D}}_t^{-1}. \end{aligned} \quad (5)$$

Here, $\boldsymbol{\Sigma}_t \in \mathbb{R}^{V \times V}$ denotes the sample covariance matrix and $\mathbf{I}_V$ and $\mathbf{1}_V$ are a $V \times V$ identity matrix and a $1 \times V$ matrix of ones, respectively. As such, $f_{\theta_N}(\cdot)$ computes a correlation matrix $\mathbf{F}_t \in [-1,1]^{V \times V}$ by normalizing $\boldsymbol{\Sigma}_t$ with the square roots of its diagonal elements $\tilde{\mathbf{D}}_t$. By constructing the dynamic node features this way, each node's feature vector has a length $B = V$. This choice of node features is motivated by previous work on static brain graphs, where a node's FC profile has shown superior performance compared to other features (Li et al., 2021; Kan et al., 2022; Cui et al., 2022).

### 4.2. Dynamic graph classifier

**Spatial-temporal feature extraction** To learn a spatial-temporal representation of the dynamic graph $G_{1:T} =$ $(\mathbf{A}_{1:T}, \mathbf{F}_{1:T})$ from (4) and (5), we employ a $L_C$-layered recurrent GNN (Seo et al., 2018), denoted $f_{\theta_F}(\cdot)$. For simplicity, we implement the recurrence relation using a gated recurrent unit (GRU) (Cho et al., 2014) and each gate as a graph convolutional network (GCN) (Kipf & Welling, 2016). Specifically, the GCN for gate $g \in \{r, u, c\}$ in the $l$-th layer is defined

$$\text{GCN}_g^{(l)}(\mathbf{F}_t, \mathbf{A}_t) = \hat{\mathbf{D}}_t^{-1/2} \hat{\mathbf{A}}_t \hat{\mathbf{D}}_t^{-1/2} \mathbf{F}_t \mathbf{W}_g^{(l)} \quad (6)$$

where $\mathbf{W}_g^{(l)} \in \mathbb{R}^{K_C \times K_C}$ is a trainable weight matrix, $\hat{\mathbf{A}}_t = \mathbf{A}_t + \mathbf{I}_V$ denotes a snapshot adjacency matrix with added self-edges, and $\hat{\mathbf{D}}_t = \text{diag}(\hat{\mathbf{A}}_t \mathbf{1}_{1 \times V}^\top)$ is the degree matrix. At each snapshot, the $l$-th layer of the GRU is defined

$$\mathbf{R}_t^{(l)} = \text{Sigmoid}\Big(\text{GCN}_r^{(l)}(\tilde{\mathbf{H}}_t^{(l-1)} || \tilde{\mathbf{H}}_{t-1}^{(l)}, \mathbf{A}_t)\Big) \quad (7)$$

$$\mathbf{U}_t^{(l)} = \text{Sigmoid}\Big(\text{GCN}_u^{(l)}(\tilde{\mathbf{H}}_t^{(l-1)} || \tilde{\mathbf{H}}_{t-1}^{(l)}, \mathbf{A}_t)\Big) \quad (8)$$

$$\mathbf{C}_t^{(l)} = \text{Tanh}\Big(\text{GCN}_c^{(l)}(\tilde{\mathbf{H}}_t^{(l-1)} || \mathbf{R}_t^{(l)} \odot \tilde{\mathbf{H}}_{t-1}^{(l)}, \mathbf{A}_t)\Big) \quad (9)$$

$$\tilde{\mathbf{H}}_t^{(l)} = \mathbf{U}_t^{(l)} \odot \tilde{\mathbf{H}}_{t-1}^{(l)} + (1 - \mathbf{U}_t^{(l)}) \odot \mathbf{C}_t^{(l)} \quad (10)$$

where $\mathbf{R}_t^{(l)}, \mathbf{U}_t^{(l)} \in \mathbb{R}^{V \times K_C}$ are the reset and update gates, respectively. The initial hidden state $\tilde{\mathbf{H}}_t^{(0)} = \mathbf{F}_t$ is set to the input node features, and $\tilde{\mathbf{H}}_0^{(l)} \in \mathbf{0}_{V \times K_C}$ is initialized as a matrix of zeros. The symbols $\odot$ and $||$ denote the Hadamard product and the feature-wise concatenation operator, respectively. By iterating through (7)-(10) for each snapshot, we obtain per-layer output embeddings $\mathbf{H}_{1:T}^{(l)} \in \mathbb{R}^{V \times K_C \times T}$. These embeddings are concatenated along the feature dimension, to combine information from neighbors that are up to $L_C$-hops away from each node, and then averaged over the node dimension to create a sequence of brain graph embeddings following

$$\mathbf{H}_{1:T}^C = \boldsymbol{\phi}(||_{l=1}^{L_C} \mathbf{H}_{1:T}^{(l)}) \in \mathbb{R}^{K_C L_C \times T} \quad (11)$$

where $\boldsymbol{\phi} = \frac{1}{V} \mathbf{1}_{1 \times V}$ represents a average pooling matrix. Though we only focus on GCN and GRU, it is straightforward to extend the proposed method to other GNNs (Hamilton et al., 2017; Veličković et al., 2017) and RNNs (Hochreiter & Schmidhuber, 1997), which we leave for future work.

**Temporal attention readout** In order to highlight the snapshots that contain the most relevant brain graph embeddings $\mathbf{H}_{1:T}^C$ from (11), we incorporate a novel temporal attention readout layer, denoted $f_{\theta_T}(\cdot)$, which is adapted from squeeze-and-excite attention networks (Hu et al., 2018). More formally, the layer introduces a temporal attention score matrix $\boldsymbol{\alpha} \in (0,1)^{1 \times T}$ defined following

$$\boldsymbol{\alpha} = \text{Sigmoid}(\text{ReLU}(\boldsymbol{\psi} \mathbf{H}_{1:T}^C \mathbf{W}_1) \mathbf{W}_2) \quad (12)$$

where $\mathbf{W}_1 \in \mathbb{R}^{T \times \tau T}$, $\mathbf{W}_2 \in \mathbb{R}^{\tau T \times T}$ are trainable weight matrices that capture temporal dependencies. The hyper-parameter $\tau \in (0,1]$ controls the bottleneck, and $\boldsymbol{\psi} = \frac{1}{K_C L_C} \mathbf{1}_{1 \times K_C L_C}$ represents an average pooling matrix.

**Graph-level representation** Using the temporal attention scores $\boldsymbol{\alpha}$ from (12), we compute the graph-level representation $\mathbf{h}_{\mathcal{G}} \in \mathbb{R}^{L_C K_C}$ using a weighted sum of the snapshots

$$\mathbf{h}_{\mathcal{G}} = (\boldsymbol{\alpha} \odot \mathbf{H}_{1:T}^C)\boldsymbol{\xi}^\top \qquad (13)$$

where $\xi = \mathbf{1}_{1 \times T}$ denotes a sum pooling matrix. Finally, we pass the graph-level representation through a linear layer, which maps it onto a vector of class probabilities $p(y|\mathbf{X}_{1:T'}) \in \Delta^C$. Formally

$$p(y|\mathbf{X}_{1:T'}) = \mathrm{Softmax}(\mathbf{h}_{\mathcal{G}}\mathbf{W}_3) \qquad (14)$$

where $\mathbf{W}_3 \in \mathbb{R}^{L_C L_K \times C}$ is a trainable weight matrix.

### 4.3. Loss function

Since our task is graph classification, we train DynDepNet by minimizing the cross-entropy loss $\mathcal{L}_{\mathrm{CE}}(y, \hat{y})$ as well as a collection of prior constraints on the learnt graphs denoted $\mathcal{R}(G_{1:T})$ such that

$$\mathcal{L}(y, \{\hat{y}, G_{1:T}\}) = \mathcal{L}_{\mathrm{CE}}(y, \hat{y}) + \mathcal{R}(G_{1:T}). \qquad (15)$$

Here, $\mathcal{L}_{\mathrm{CE}}(y, \hat{y}) = -\sum_{c=1}^{C} \mathbb{1}(y = c) \log p(y|\mathbf{X}_{1:T'})_c$ and $p(\cdot|\cdot)_c$ is the $c$-th element of the output probability vector from (14). This encourages DynDepNet to learn task-aware dynamic graphs that encode class differences into $\mathbf{A}_{1:T}$.

**Regularization constraints** Since connected nodes in a graph are more likely to share similar features (McPherson et al., 2001), we add a regularization term encouraging feature smoothness defined as

$$\mathcal{L}_{\mathrm{FS}}(\mathbf{A}_{1:T}, \mathbf{F}_{1:T}) = \frac{1}{V^2} \sum_{t=1}^{T} \mathrm{Tr}(\mathbf{F}_t^\top \hat{\mathbf{L}}_t \mathbf{F}_t) \qquad (16)$$

where $\mathrm{Tr}(\cdot)$ denotes the matrix trace operator and $\hat{\mathbf{L}}_t = \mathbf{D}_t^{-1/2}\mathbf{L}_t\mathbf{D}_t^{-1/2}$ is the (symmetric) normalized Laplacian matrix defined as $\mathbf{L}_t = \mathbf{D}_t - \mathbf{A}_t$ where $\mathbf{D}_t = \mathrm{diag}(\mathbf{A}_t\mathbf{1}_{V \times 1})$ which makes feature smoothness node degree independent (Ando & Zhang, 2006). Furthermore, to discourage volatile changes in graph structure between snapshots we also add a prior constraint encouraging temporal smoothness defined as

$$\mathcal{L}_{\mathrm{TS}}(\mathbf{A}_{1:T}) = \sum_{t=1}^{T-1} ||\mathbf{A}_t - \mathbf{A}_{t+1}||_1 \qquad (17)$$

where $|| \cdot ||_1$ denotes the matrix L1-norm. Moreover, to encourage the learning of a large sparsity parameter $\mathrm{Softmax}(\theta_P)$ in (4), we further add a sparsity regularization term defined

$$\mathcal{L}_{\mathrm{SP}}(\mathbf{A}_{1:T}) = \sum_{t=1}^{T} ||\mathbf{A}_t||_1, \qquad (18)$$

which in combination with $\mathcal{L}_{\mathrm{CE}}(\cdot, \cdot)$, ensures only the most import task-specific edges are kept in $\mathbf{A}_t$. The final loss function we seek to minimize is

$$\mathcal{L}(y, \{\hat{y}, G_{1:T}\}) = \mathcal{L}_{\mathrm{CE}}(y, \hat{y}) + \lambda_{\mathrm{FS}}\mathcal{L}_{\mathrm{FS}}(\mathbf{F}_{1:T}, \mathbf{A}_{1:T})$$
$$+ \lambda_{\mathrm{TS}}\mathcal{L}_{\mathrm{TS}}(\mathbf{A}_{1:T}) + \lambda_{\mathrm{SP}}\mathcal{L}_{\mathrm{SP}}(\mathbf{A}_{1:T}) \quad (19)$$

where $\lambda_{\mathrm{FS}}, \lambda_{\mathrm{TS}}, \lambda_{\mathrm{SP}} \geq 0$ are hyperparameters weighting the relative contribution of each regularization term.

## 5. Experiments

We evaluate the performance of DynDepNet on the task of biological sex classification, which serves as a widely used benchmark for supervised deep learning models designed for fMRI data (Kim et al., 2021; Azevedo et al., 2022). The classification of biological sex based on brain imaging data is of significant interest in neuroscience, as it has been shown that there are observable differences between males and females in various brain characteristics and functional connectivity patterns (Bell et al., 2006; Mao et al., 2017). By assessing the performance of DynDepNet on this task, we can gain insights into its ability to capture sex-related differences and leverage them for accurate classification.

### 5.1. Datasets

We constructed two datasets using publicly available fMRI data from the Human Connectome Project (HCP) S1200 release[2] (Van Essen et al., 2013). The two datasets differ based on whether the subjects were in resting state (HCP-Rest) or performing a specific task (HCP-Task) during the fMRI acquisition.

**HCP-Rest** For the HCP-Rest, we considered resting-state fMRI scans that underwent minimal preprocessing following the pipeline described in Glasser et al. (2013). We selected a total of $N = 1,095$ subjects from the first scanning-session (out of four) that used left-right (LR) phase encoding. During image acquisition, subjects were instructed to rest for 15 minutes. The repetition time (TR), which denotes the time between successive image acquisitions, was set to $0.72$ seconds, resulting in a total of $T' = 1,200$ images per subject. We used the biological sex of each subject as the label resulting in a total of $C = 2$ classes. Female subjects accounted for $54.4\%$ of the dataset.

**HCP-Task** For the HCP-Task, we considered task fMRI scans from the emotional task, which underwent minimal preprocessing using the same pipeline as HCP-Rest. We selected a total of $N = 926$ subjects from the first scanning session (out of two) with LR phase encoding. During the task, subjects were asked to indicate which of two faces or

---

[2]https://db.humanconnectome.org

Table 1. Biological sex classification results for HCP-Rest and HCP-Task. Results are mean plus/minus standard deviation across 5 runs. First and second-best results are **bold** and underlined, respectively. Statistically significant difference from DynDepNet marked *. Models are grouped by type of functional connectivity taken as input (FC) (S = static, D = dynamic, - = BOLD signals).

| Model | FC | HCP-Rest | | HCP-Task | |
|---|---|---|---|---|---|
| | | ACC (%, ↑) | AUROC (↑) | ACC (%, ↑) | AUROC (↑) |
| BLSTM | - | $81.50 \pm 1.26$ * | $0.9058 \pm 0.0081$ * | $77.24 \pm 4.05$ * | $0.8526 \pm 0.0188$ * |
| KRR | S | $83.50 \pm 1.94$ * | $0.9187 \pm 0.0025$ * | $81.37 \pm 2.17$ * | $0.9031 \pm 0.0185$ * |
| SVM | S | $82.70 \pm 2.68$ * | $0.9170 \pm 0.0089$ * | $83.16 \pm 1.91$ * | $0.9097 \pm 0.0184$ * |
| MLP | S | $81.47 \pm 3.29$ * | $0.9091 \pm 0.0281$ * | $81.10 \pm 3.44$ * | $0.8837 \pm 0.0250$ * |
| BNCNN | S | $76.83 \pm 7.46$ * | $0.6156 \pm 0.0837$ * | $70.66 \pm 8.23$ * | $0.5945 \pm 0.0499$ * |
| DFMRI | S | $82.65 \pm 3.40$ * | $0.8941 \pm 0.0342$ * | $81.34 \pm 2.19$ * | $0.8024 \pm 0.0317$ * |
| FBNG | S | $81.57 \pm 2.90$ * | $0.8967 \pm 0.0170$ * | $77.16 \pm 3.90$ * | $0.8548 \pm 0.0320$ * |
| STGCN | D | $62.63 \pm 4.50$ * | $0.6991 \pm 0.0264$ * | $54.87 \pm 3.37$ * | $0.5629 \pm 0.0355$ * |
| STAGIN | D | $83.13 \pm 2.11$ * | $0.8597 \pm 0.0467$ * | $81.88 \pm 2.73$ * | $0.8088 \pm 0.0404$ * |
| DynDepNet | D | $\mathbf{92.32 \pm 2.22}$ | $\mathbf{0.9623 \pm 0.0433}$ | $\mathbf{89.54 \pm 3.48}$ | $\mathbf{0.9496 \pm 0.0423}$ |

which of two shapes presented at the bottom of a screen matched the face or shape at the top of the screen. The scanning session lasted approximately $2.11$ minutes with a TR of $0.72$ seconds, resulting in $T' = 176$ images per subject. Similar to the HCP-Rest dataset, the classes were defined based on the biological sex of the subjects. Female subjects accounted for $51.2\%$ of the dataset.

**Further preprocessing**   Since the fMRI scans from both datasets represent a timeseries of 3D brain volumes, we parcellate them into $V = 243$ mean brain region (210 cortical, 36 subcortical) BOLD signals of length $T'$ time points using the Brainnetome atlas [3] (Fan et al., 2016). Each timeseries was then transformed into a $z$-scores by standardizing region-wise in order to remove amplitude effects. To balance the class proportions across both datasets, we randomly oversampled the minority class male.

### 5.2. Baselines

We compared DynDepNet against a range of different baseline models that have been previously used to classify fMRI data, and for which code is publicly available. The baselines are broadly grouped based on whether they take as input static FC, dynamic FC, or BOLD signals. For static (linear) baselines, we include kernel ridge regression (KRR) (He et al., 2020) and support vector machine (SVM) (Abraham et al., 2017). For static deep learning baselines we include a multilayer perception (MLP) and BrainnetCNN (BNCNN) (Kawahara et al., 2017). For dynamic baselines we include ST-GCN (STGCN) (Gadgil et al., 2020) and STAGIN (Kim et al., 2021). For GSL baselines, we include FBNetGen (FBNG) (Kan et al., 2022) and Deep fMRI (DFMRI) (Riaz et al., 2020). Finally, we include a bidirec-

tional LSTM (BLSTM) (Hebling Vieira et al., 2021) which learns directly from BOLD signals. Further details about each baseline model can be found in Appendix A.

### 5.3. Evaluation metrics

Model performance is evaluated on test data using accuracy (ACC) and area under the receiver operator curve (AUROC), as classes are balanced. To compare the performance of models, we use the almost stochastic order (ASO) test of statistical significance (Del Barrio et al., 2018; Dror et al., 2019), as implemented by Ulmer et al. (2022). For all tests, the significance level is set to $\alpha = 0.05$ and adjusted using the Bonferroni correction (Bonferroni, 1936) when making multiple comparisons.

### 5.4. Implementation

Both datasets were split into training, validation, and test datasets, maintaining class proportions with an 80/10/10% ratio. To ensure that differences in classification performance could be attributed primarily to differences in model architecture, specific training and testing strategies for the baselines were removed. Instead, all models were trained using the Adam optimizer (Kingma & Ba, 2014) with decoupled weight decay (Loshchilov & Hutter, 2017). We trained all models for a maximum of $5,000$ epochs and employed early stopping with a patience of 15 based on the lowest accuracy observed on the validation dataset. This strategy helped prevent overfitting and allowed us to select the best-performing model. To alleviate computational load and introduce stochastic augmentation, the time dimension of the training samples was randomly sampled. Specifically, for the HCP-Rest dataset, the time dimension was set to $T' = 600$, while for the HCP-Task dataset, it was set to $T' = 150$, following the approach used by Kim et al.

---

[3] https://atlas.brainnetome.org

(2021). To account for variability in the training process, each model was trained five times using different random seeds and dataset splits. This ensured a more robust evaluation of the model's performance and mitigated the influence of random initialization and data variability.

**Software and hardware** All models were developed in Python 3.7 (Python Core Team, 2019) and relied on several libraries including scikit-learn 1.3.0 (Pedregosa et al., 2011), PyTorch 2.0.1 (Paszke et al., 2019), numpy 1.25.1 (Harris et al., 2020), TorchMetrics 1.0.0 (Nicki Skafte Detlefsen et al., 2022), and deep-significance 1.2.6 (Ulmer et al., 2022). All experiments were conducted on a Linux server (Debian 5.10.113-1) with a NVIDIA RTX A6000 GPU with 48 GB memory and 16 CPUs.

**Hyperparameter optimization** In order to determine the best hyperparameter values for each model, we performed a grid search using the validation dataset. We started with the default hyperparameter values provided in the original implementations of the baselines. While it was not feasible to exhaustively search for the optimal values of all hyperparameters for every baseline model, we focused on tuning key hyperparameters such as regularization loss weights (for KRR and SVM) and the dimensions of hidden layers (for MLP, BLSTM, BNCNN, STGCN, DFMRI, FBNG, STAGIN). For DynDepNet, we set the number of filters in the temporal feature extractor $f_{\theta_E}(\cdot)$ to $M = 3$ and the bottleneck in the temporal attention layers $f_{\theta_T}(\cdot)$ to $\tau = 0.5$. The specific values of other hyperparameters used for DynDepNet can be found in Appendix B.

## 5.5. Classification results

The results of biological sex classification are presented in Table 1. It is evident that DynDepNet achieves the highest performance among all models on both HCP-Rest and HCP-Task datasets, as measured by accuracy and AUROC. On HCP-Rest, DynDepNet surpasses the second-best baseline KRR in terms of accuracy by 8.82 percentage points (pp), while on HCP-Task, it outperforms the second-best baseline SVM by 8.17 pp. These improvements are statistically significant. It is worth noting that, consistent with the findings of He et al. (2020), the linear baselines KRR and SVM either outperform or achieve comparable performance to the deep learning baselines when considering only static brain graphs. The superior performance of DynDepNet can be attributed to its ability to learn dynamic brain graphs during training, as well as incorporating temporal dynamics. See Appendix C for further results.

## 5.6. Ablation study

To examine the impact of key components in DynDepNet, we conduct an ablation study by removing specific model components and evaluating their effect on performance. Specifically, within the dynamic graph learner $F_{\theta_G}(\cdot)$ we replace the I-TCN $f_{\theta_E}(\cdot)$ with a 1D CNN with filter length 4 (w/o inception), replace self-attention $f_{\theta_S}(\cdot)$ with a normalized Person correlation matrix (w/o spatial att.), remove edge sparsity $f_{\theta_P}(\cdot)$ with $\lambda_{SP} = 0$ (w/o sparsity), and remove temporal attention $f_{\theta_T}(\cdot)$ (w/o temporal att.). In addition, we also remove feature smoothness $\lambda_{FS} = 0$ (w/o feature reg.) and temporal smoothness $\lambda_{TS} = 0$ (w/o temporal reg.) graph regularization terms from the loss function.

Table 2. Ablation study results on HCP-Rest and HCP-Task. Results are mean plus/minus standard deviation across 5 runs. Best results are shown in **bold**. Statistically significant difference from DynDepNet marked *.

| Model | ACC (%, ↑) | |
| --- | --- | --- |
| | **HCP-Rest** | **HCP-Task** |
| DynDepNet | **92.32 ± 2.22** | **89.54 ± 3.48** |
| - w/o inception | 91.22 ± 2.69 * | 88.79 ± 2.34 |
| - w/o self att. | 89.97 ± 3.04 * | 86.80 ± 3.76 |
| - w/o sparsity | 92.32 ± 2.23 * | 87.00 ± 2.33 * |
| - w/o temporal att. | 92.26 ± 2.43 | 87.56 ± 2.84 * |
| - w/o feature reg. | 92.29 ± 2.39 | 88.50 ± 2.87 |
| - w/o temporal reg. | 92.12 ± 2.21 | 88.43 ± 3.23 |

**Results** The results of the ablation study are presented in Table 2. First, the use of self-attention significantly improves accuracy across both datasets (HCP-Rest ↑ 2.34 pp vs HCP-Task ↑ 2.74 pp) since it allows for task-aware spatial relationships between brain regions to be built into the dynamic graph adjacency matrix, which in turn benefits the graph classifier. Second, sparsity also significantly improves accuracy (HCP-Rest ↑ 1.04 pp vs HCP-Task ↑ 2.54 pp) since it removes noisy edges from the dynamic adjacency matrix thereby reducing errors from being propagated to node representations in the graph classifier. Finally, the effect of the I-TCN is significant for HCP-Rest (↑ 1.10 pp) but only marginal for HCP-Task (↑ 0.75 pp). This might be explained by the fact that the BOLD signals from the former dataset are collected over a longer time period than the latter, thereby benefiting more from larger kernel sizes being able to extract longer temporal patterns.

## 5.7. Hyperparameter sensitivity

We conduct a sensitivity analysis on the main hyperparameters which influence the complexity of DynDepNet. In particular, for the dynamic graph learner $F_{\theta_G}(\cdot)$ we vary window length $P$, window stride $S$, embedding size $K_E$, and number of layers $L_G$. On the other hand, for the dynamic graph classifier $F_{\theta_C}(\cdot)$ we vary the number of layers

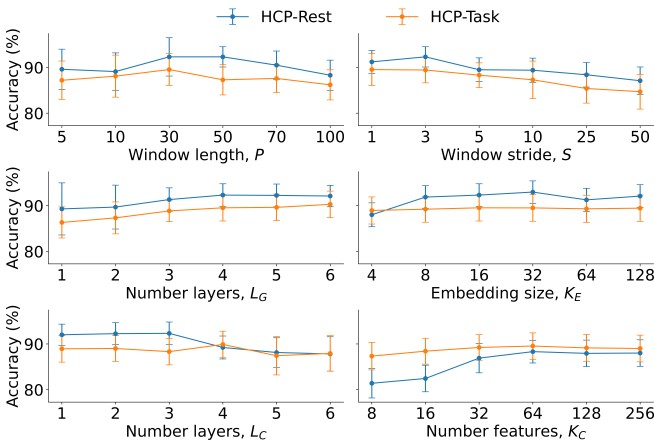

*Figure 2.* Sensitivity analysis results on HCP-Rest and HCP-Task. Results are mean plus/minus standard deviation across 5 runs.

$L_C$ and number of features $K_C$. For each experiment, we change the hyperparameter under investigation and fix the remaining hyperparameters to their optimally tuned values from Appendix B.

**Results** Figure 2 summarizes the results of the hyperparameter sensitivity analysis. On both HCP-Rest and HCP-Task we observe that increasing the window length $P$ leads to a decrease in accuracy. We attribute this to the fact that including more data within a window makes it harder for the dynamic graph learner to identify fast changes between brain regions that are discriminative for the task. Similarly, increasing the window stride $S$ also leads to a decrease in accuracy, as contiguous windows of BOLD signals are not fully captured, resulting in a loss of important information when constructing dynamic graphs. Furthermore, we find that increasing the depth of information propagation beyond 3 hops in the dynamic graph classifier, as indicated by the number of layers $L_C$, leads to a decrease in performance. This suggests that excessive information propagation can introduce noise and degrade the discriminative power of the learned graph representations. Moreover, increasing the number of layers $L_G$ and the embedding size $K_E$ in the graph learner, as well as the number of features $K_C$ in the graph classifier, exhibit diminishing returns in terms of performance gains. This implies that there is an optimal balance between model complexity and performance, beyond which further increasing the capacity of the model does not yield significant improvements in accuracy.

## 6. Interpretability analysis

A major strength of DynDepNet is its ability to learn task-aware dynamic brain graphs from BOLD signals, which has applications beyond classification. To showcase the in-

terpretability of DynDepNet, Figure 3 compares, using the same subject, a dynamic adjacency matrix $\mathbf{A}_{1:T}$ output from DynDepNet with a dynamic FC matrix $\mathbf{A}^C_{1:T}$ calculated using Pearson correlation following Calhoun et al. (2014). The comparison highlights several advantages of the learned dynamic adjacency matrix. Firstly, $\mathbf{A}_{1:T}$ exhibits greater sparsity, with only the most relevant and discriminative connections between brain regions having non-zero weights. In contrast, $\mathbf{A}^C_{1:T}$ assigns weights to all connections, including potentially irrelevant and noisy relationships. Additionally, the learned $\mathbf{A}_{1:T}$ is not restricted to linear relationships between brain regions, as it can capture more complex and non-linear dependencies. This flexibility allows DynDepNet to capture higher-order interactions and intricate dynamics that may be missed by $\mathbf{A}^C 1:T$, which assumes linear associations between brain regions.

### 6.1. Brain region importance

To identify brain regions that are most sex-discriminative, we create a brain region score vector $\mathbf{z} \in \mathbb{R}^V$ by computing the temporally weighted node degree. This score vector provides a measure of the importance of each brain region in contributing to the classification task. Specifically, we calculate the average degree for each region across snapshots, weighted by the corresponding temporal attention scores $\boldsymbol{\alpha}$. This computation yields $\mathbf{z} = \frac{1}{T}\sum_{t=1}^{T}(\sum_{j=1}^{V}\mathbf{A}_{j,t})\alpha_t$. Next, we select the top 20% of regions based on their scores across all subjects in the test dataset. These regions are then visualized with respect to the functional connectivity networks defined by Yeo et al. (2011) in Figure 4.

**Results** For HCP-Rest, 25.5% of the highest scoring brain regions are within the default mode network (DMN), a key FC network that is consistently observed in resting-state fMRI studies (Mak et al., 2017; Satterthwaite et al., 2015). Within the DMN the brain regions with the highest predictive ability are localized in the dorsal anterior cingulate cortex, middle frontal gyrus, and posterior superior temporal cortex. These fronto-temporal brain regions are key components of the theory of mind network, which underlies a meta-cognitive function in which females excel (Adenzato et al., 2017). Another key region in theory of mind tasks, the posterior superior temporal cortex is found to reliably predict sex within the ventral attention network (VAN). For HCP-Task, 30.6% of the highest scoring brain regions fall in the parahippocampal gyrus, medial occipital cortex, and superior parietal lobule which form the posterior visual network (VSN). The fact that such regions best discriminated males from females reflects differences in the ability to process emotional content and/or sex-related variability in directing attention to certain features of emotional stimuli (Mackiewicz et al., 2006), like the facial expressions from the HCP task paradigm (Markett et al., 2020). For

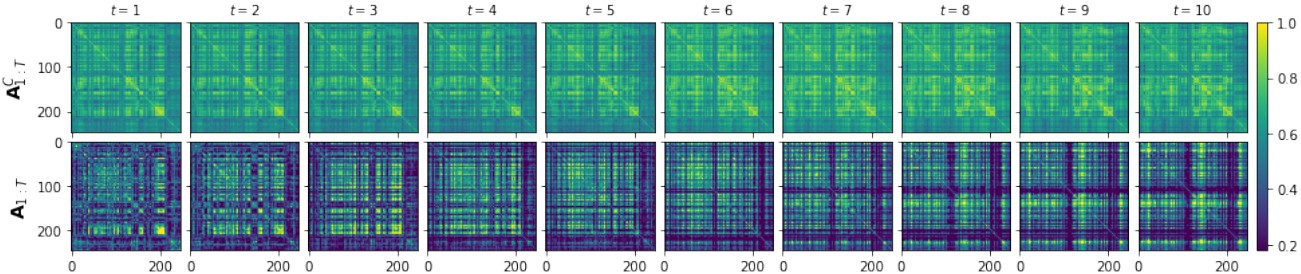

*Figure 3.* Dynamic FC matrix calculated using Pearson correlation $\mathbf{A}^{C}_{1:T}$ normalized to $[0, 1]$ (top) compared to a dynamic adjacency matrix learnt by DynDepNet $\mathbf{A}_{1:T}$ (bottom). Both matrices are computed using the BOLD signals from the same randomly sampled subject from HCP-Rest with a window size and stride of $P = 50$ and $S = 3$, respectively.

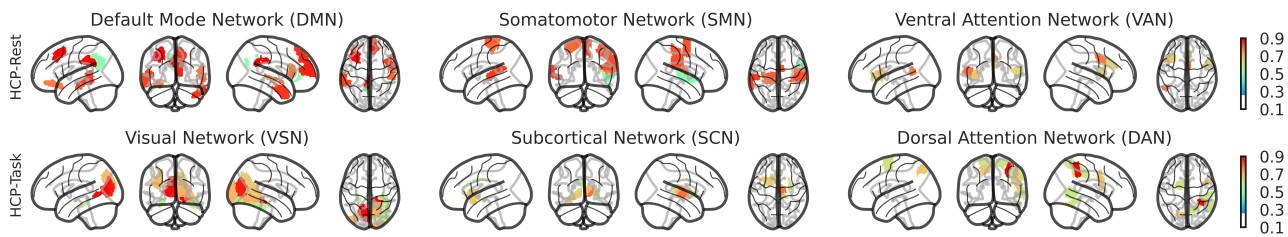

*Figure 4.* Sex-discriminative brain region scores $\mathbf{z}$ (normalized to $[0, 1]$) for HCP-Rest (top) and HCP-Task (bottom).

further analysis we refer to Appendix D.

# 7. Conclusion

In conclusion, we propose DynDepNet, a novel end-to-end trainable model for learning optimal time-varying dependency structures from fMRI data in the form of dynamic brain graphs. To the best of our knowledge, we are the first to propose and address the dynamic graph structure learning problem in the context of fMRI data using a GNN-based deep learning method. Our approach leverages spatial-temporal attention mechanisms to capture the inter- and intra-relationships of brain region BOLD signals. Through extensive experiments on two real-world fMRI datasets, we demonstrate that DynDepNet achieves state-of-the-art performance in biological sex classification. The interpretability of DynDepNet is a major strength, as it learns task-aware dynamic graphs that capture the most relevant and discriminative connections between brain regions. The learned dynamic adjacency matrix exhibits sparsity, highlighting the importance of only the most informative edges, while also allowing for non-linear relationships between brain regions. This flexibility enables DynDepNet to capture higher-order interactions and intricate dynamics that may be missed by traditional correlation-based methods.

**Future research** Future research directions will focus on learning the optimal window size and stride during training to further enhance the flexibility and adaptability of DynDepNet. Additionally, it will be valuable to expand the range of fMRI datasets and prediction tasks to include mental health disorders such as schizophrenia, depression, and autism spectrum disorders. By applying DynDepNet to these domains, possible insights might be gained into the dynamic brain connectivity patterns associated with these different psychiatric conditions, contributing to the development of more accurate diagnostic tools and personalized treatment strategies.

# Acknowledgments

We would like to thank the anonymous reviewers for their insightful comments and valuable feedback, which greatly contributed to improving the quality of this work. This research was supported by The Alan Turing Institute under the EPSRC grant EP/N510129/1. The data was provided [in part] by the Human Connectome Project, WU-Minn Consortium (Principal Investigators: David Van Essen and Kamil Ugurbil; 1U54MH091657) funded by the 16 NIH Institutes and Centers that support the NIH Blueprint for Neuroscience Research; and by the McDonnell Center for Systems Neuroscience at Washington University.

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

# A. Baselines

We compare DynDepNet against a range of different baseline models that have been previously used to classify fMRI data and for which code is publicly available. These baselines encompass a mixture of models covering both static and dynamic, linear and non-linear, as well as graph and non-graph-based. It is worth noting that many existing GNN baselines are challenging to adapt for use with fMRI data, as the data is naturally multi-graph-based (each subject has their own dynamic graph) rather than a single static or dynamic graph.

**Kernel ridge regression**[4] **(KRR)** (He et al., 2020)    Kernel ridge regression combines ridge regression (linear least squares with L2-norm regularization) with the kernel trick. It learns a linear decision function in a higher-dimensional feature space induced by the kernel and the input data (Murphy, 2012a). Following He et al. (2020), we use a linear kernel and keep the weight on the regularization loss as a tunable hyperparameter. KRR takes as input the vectorized lower-triangle (excluding the principal diagonal) of a static FC matrix computed using Pearson correlation.

**Support vector machine (SVM)** (Abraham et al., 2017)    Support vector machine learns a hyperplane to separate data points from different classes in a high-dimensional space created by a kernel function (Murphy, 2012b). Following Abraham et al. (2017), we use a linear kernel and keep the weight on the regularization loss as a tunable hyperparameter. Similar to KRR, SVM takes as input a vectorized static FC matrix computed using Pearson correlation.

**Multilayered perceptron**[4] **(MLP)** (Hebling Vieira et al., 2021)    A multilayered perceptron taking as input vectorized static FC matrices computed using Pearson correlation. This baseline model has been used in previous studies (Kawahara et al., 2017; Gadgil et al., 2020), and we follow the implementation described in Hebling Vieira et al. (2021). The MLP consists of three linear layers with dropout, batch normalization, and ReLU activation functions applied after the first two layers. The dimensionality of the hidden layers is considered as a hyperparameter to be tuned.

**Bi-directional long short-term memory (BLSTM)** (Hebling Vieira et al., 2021)    A bi-directional LSTM that directly learns patterns from the BOLD signals, without relying on precomputed FC matrices. In accordance with Hebling Vieira et al. (2021), our implementation includes two bi-directional LSTM layers (Graves & Schmidhuber, 2005). Each layer processes the BOLD signals in both the forward and backward directions, and the hidden representations from the two directions are combined using addition. The dimensionality of the hidden layers is treated as a tunable hyperparameter.

**BrainNetCNN**[4] **(BNCNN)** (Kawahara et al., 2017)    A CNN that utilizes specially designed cross convolutional filters, including edge-to-edge and edge-to-node filters, to directly learn topological features from static FC matrices. The model takes the FC matrices as input and was originally proposed in Kawahara et al. (2017). In our implementation, we follow the approach described in He et al. (2020), which includes four layers. The number of hidden channels in the last layer is treated as a tunable hyperparameter.

**Spatio-temporal graph convolutional network**[5] **(STGCN)** (Gadgil et al., 2020)    A GNN architecture that consists of three spatio-temporal blocks. Each block comprises a GCN layer for extracting spatial features and a 1D CNN layer for extracting temporal features. In our implementation, we follow the approach described in Gadgil et al. (2020). The node features are obtained by windowing the BOLD signals, and the adjacency matrix is computed as the average FC matrix, which is calculated using Pearson correlation, over all subjects in the training dataset. The number of hidden features in the model is treated as a hyperparameter to be tuned.

**Deep fMRI (DFMRI)** (Riaz et al., 2020)    A deep learning-based GSL method that directly learns static brain graphs from BOLD signals. The model consists of a 1D CNN feature extractor, an MLP graph constructor, and an MLP graph classifier. The feature extractor processes the BOLD signals to extract informative features. The graph constructor, inspired by Siamese networks (Bromley et al., 1993), learns a similarity score between pairs of extracted features from different brain regions. Finally, the graph classifier uses the learned graph structure to perform classification. In our implementation, we treat the hidden dimension in the graph classifier as a tunable hyperparameter.

---

[4]https://github.com/ThomasYeoLab/CBIG/blob/master/stable_projects/predict_phenotypes/He2019_KRDNN/

[5]https://github.com/sgadgil6/cnslab_fmri

**Functional Brain Network Generator**[6]**(FBNG)** (Kan et al., 2022)  Another GSL method that learns static brain graphs directly from BOLD signals. It utilizes a LSTM feature extractor and a GNN as a graph classifier (Kan et al., 2022). In contrast to DFMRI, which uses a MLP to learn a graph adjacency matrix, FBNG takes the inner product between the extracted features. Similar to DynDepNet, FBNG introduces a group inter loss, which aims to maximize the difference in learned graphs across different classes while keeping those within the same class similar. In our implementation, we treat the hidden dimension in the graph classifier as a tunable hyperparameter.

**Spatio-temporal Attention Graph Isomorphism Network**[7] **(STAGIN)** (Kan et al., 2022)  A joint GNN and transformer model that takes attributed unweighted dynamic graphs derived from sliding window functional connectivity as input. Following the approach of Kim et al. (2021), Pearson correlation is used as the measure of functional connectivity, and the matrices are binarized by thresholding the top 30-percentile values as connected edges. In our implementation, we fix the number of layers in the GNN to four and treat the node embedding dimension as a hyperparameter to be tuned.

## B. Hyperparameter optimization

*Table 3.* Optimal hyperparameter values for DynDepNet on HCP-Rest and HCP-Task based on 5 runs using lowest validation accuracy.

| Hyperparameter | Range | HCP-Rest | HCP-Task |
|---|---|---|---|
| Training | | | |
| - Batch size | {5, 10, 20, 50} | 20 | 20 |
| - Learning rate | {1e-2, 1e-3, 1e-4} | 1e-3 | 1e-3 |
| - Weight decay | {1e-5, 1e-4, 1e-3} | 1e-4 | 1e-4 |
| Model | | | |
| - Dynamic graph learner | | | |
|   – Window length, $P$ | {5, 10, 30, 50, 70, 100} | 50 | 30 |
|   – Window stride, $S$ | {1, 3, 5, 10, 25, 50} | 3 | 1 |
|   – Number of layers, $L_G$ | {1, 2, 3, 4, 5, 6} | 4 | 4 |
|   – Number of features, $K_E$ | {8, 16, 32, 128, 256} | 64 | 64 |
|   – Filter sizes, $S_m$ | {{3, 5, 7}, {4, 8, 16}} | {4, 8, 16} | {4, 8, 16} |
|   – Embedding size $K_S$ | {4, 8, 16, 32, 64, 128} | 16 | 16 |
| - Dynamic graph classifier | | | |
|   – Number of layers, $L_C$ | {1, 2, 3, 4, 5, 6} | 3 | 3 |
|   – Number of features, $K_C$ | {8, 16, 32, 64, 128, 256} | 64 | 64 |
| - Feature smoothness, $\lambda_{\text{FS}}$ | {1e-4, 1e-3, 1e-2} | 1e-4 | 1e-4 |
| - Temporal smoothness, $\lambda_{\text{TS}}$ | {1e-4, 1e-3, 1e-2} | 1e-3 | 1e-4 |
| - Sparsity, $\lambda_{\text{SP}}$ | {1e-4, 1e-3, 1e-2} | 1e-3 | 1e-3 |

## C. Classification results

Figure 5 shows individual ASO test (Dror et al., 2019) statistics for the biological sex classification task. The ASO test has been recently proposed to test the statistical significance of deep learning models. Specifically, the ASO test determines whether a stochastic order (Reimers & Gurevych, 2018) exists between two models based on their respective sets of scores obtained from multiple runs using different random seeds. Given scores of two models $A$ and $B$ over multiple runs, the ASO test computes a test-statistic $\epsilon_{\min}$ that indicates how far model $A$ is from being significantly better than model $B$. Given a predefined significance level $\alpha \in (0, 1)$, when the distance $\epsilon_{\min} = 0.0$, one can claim that model $A$ is stochastically dominant over model $B$. When $\epsilon_{\min} < 0.5$ one can say model $A$ almost stochastically dominates model $B$. Finally, when $\epsilon_{\min} = 1.0$ model $B$ stochastically dominates model $A$. For $\epsilon_{\min} = 0.5$, no order between model $A$ and model $B$ can be determined.

[6] https://github.com/Wayfear/FBNETGEN
[7] https://github.com/egyptdj/stagin

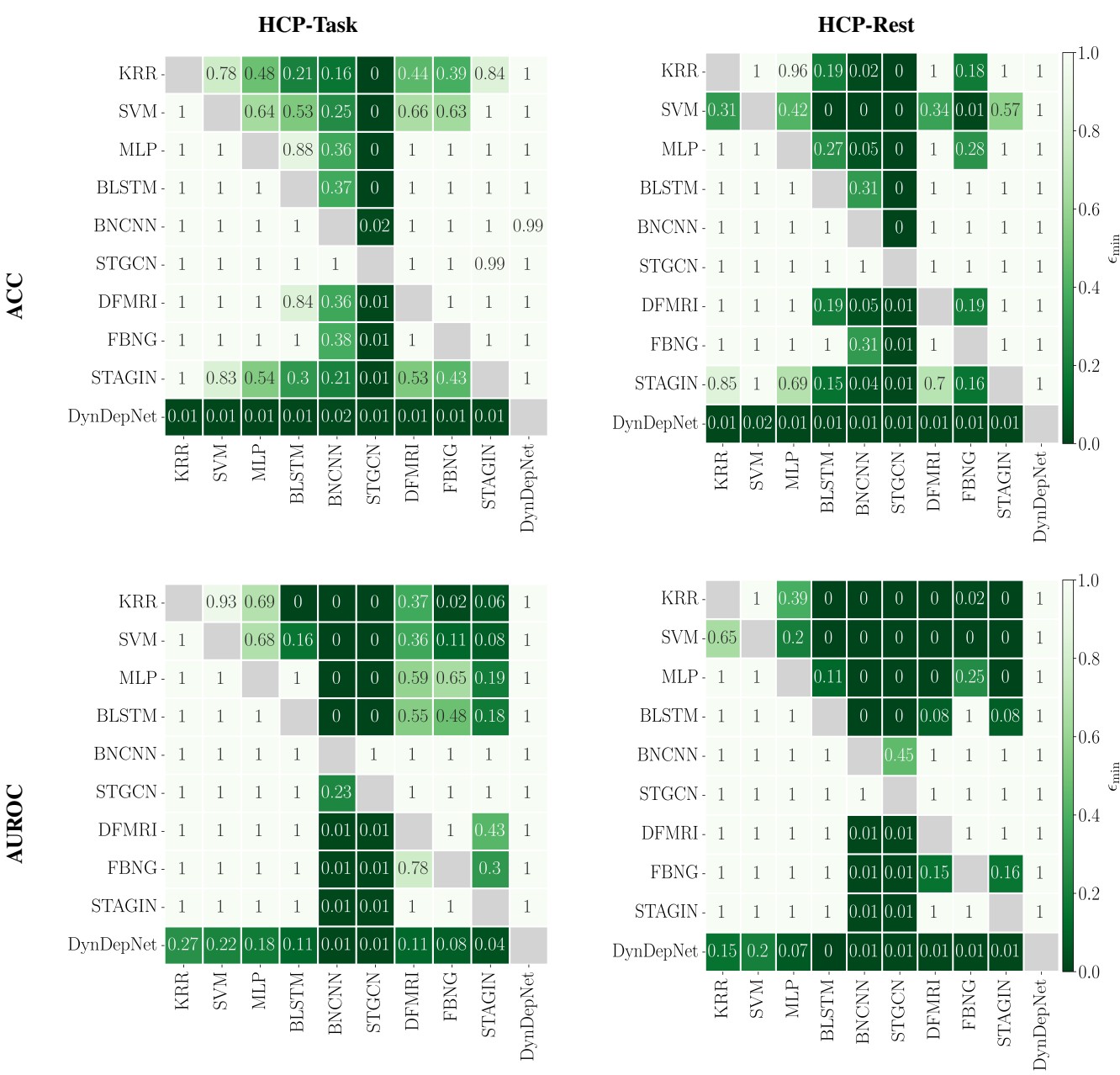

*Figure 5.* ASO test statistics $\epsilon_{\min}$ for biological sex classification on HCP-Rest and HCP-Task using significance level $\alpha = 0.05$. Results are read from row to column. For example, on HCP-Rest in terms of accuracy (top left) DynDepNet (row) is stochastically dominant over STAGIN (column) with $\epsilon_{\min}$ of 0.01.

## D. Brain region importance

We provide further details on the sex-discriminative brain region scores from Figure 4 in Tables 4-9. All brain regions and their respective MNI coordinates are taken from the Brainnetome atlas (Fan et al., 2016). Brain regions are further grouped into FC networks from Yeo et al. (2011) as well as lobe and gyrus (the outermost layer of the brain).

*Table 4.* Sex-discriminative brain region scores (normalized to [0, 1]) the default mode network (DMN) for HCP-Rest (Figure 4 top left).

| Lobe | Gyrus (hemisphere) | Region | MNI (x, y, z) | Score |
|---|---|---|---|---|
| Limbic lobe | Cingulate gyrus (left) | A23d dorsal area 23 | -4, -39, 31 | 0.93 |
| Frontal lobe | Middle frontal gyrus (left) | A8vl ventrolateral area 8 | -33, 23, 45 | 0.92 |
| Frontal lobe | Superior frontal gyrus (right) | A10m medial area 10 | 8, 58, 13 | 0.87 |
| Frontal lobe | Superior frontal gyrus (right) | A9l lateral area 9 | 13, 48, 40 | 0.85 |
| Temporal lobe | Middle temporal gyrus (right) | A21r rostral area 21 | 51, 6, -32 | 0.83 |
| Temporal lobe | Posterior superior temporal sulcus (left) | rpSTS rostroposterior superior temporal sulcus | -54, -40, 4 | 0.82 |
| Temporal lobe | Middle temporal gyrus (left) | A21c caudal area 21 | -65, -30, -12 | 0.81 |
| Temporal lobe | Superior temporal gyrus (right) | A22r rostral area 22 | 56, -12, - 5 | 0.72 |
| Frontal lobe | Inferior frontal gyrus (right) | A45c caudal area 45 | 54, 24, 12 | 0.72 |
| Frontal lobe | Orbital gyrus (left) | A12/47o orbital area 12/47 | -36, 33, -16 | 0.69 |
| Parietal lobe | Precuneus (left) | A31 Area 31 (Lc1) | -6, -55, 34 | 0.23 |
| Limbic lobe | Cingulate gyrus (right) | A32sg subgenual area 32 | 5, 41, 6 | 0.23 |

*Table 5.* Sex-discriminative brain region scores (normalized to [0, 1]) in the somatomotor network (SMN) for HCP-Rest (Figure 4 top middle).

| Lobe | Gyrus (hemisphere) | Region | MNI (x, y, z) | Score |
|---|---|---|---|---|
| Temporal lobe | Superior temporal gyrus (left) | A22c caudal area 22 | -62, -33, 7 | 0.83 |
| Parietal lobe | Inferior parietal lobule (right) | A40rv rostroventral area 40 (PFop) | 55, -26, 26 | 0.79 |
| Frontal lobe | Superior frontal gyrus (right) | A6m medial area 6 | 7, -4, 60 | 0.78 |
| Parietal lobe | Postcentral gyrus (right) | A2 area 2 | 48, -24, 48 | 0.77 |
| Frontal lobe | Precentral gyrus (left) | A4t area 4 (trunk region) | -13, -20, 73 | 0.77 |
| Frontal lobe | Precentral gyrus (left) | A4ul area 4 (upper limb region) | -26, -25, 63 | 0.73 |
| Parietal lobe | Postcentral gyrus (left) | A1/2/3tru area 1/2/3 (trunk region) | -21, -35, 68 | 0.71 |
| Parietal lobe | Postcentral gyrus (right) | A1/2/3tonIa area 1/2/3 (tongue and larynx region) | 56, -10, 15 | 0.70 |
| Temporal lobe | Superior temporal gyrus (right) | TE1.0 and TE1.2 | 51, -4, -1 | 0.22 |

*Table 6.* Sex-discriminative brain region scores (normalized to [0, 1]) in the ventral attention network (VAN) for HCP-Rest (Figure 4 top right).

| Lobe | Gyrus (hemisphere) | Region | MNI (x, y, z) | Score |
|---|---|---|---|---|
| Temporal lobe | Posterior superior temporal sulcus (left) | Caudoposterior superior temporal sulcus | -52, -50, 11 | 0.72 |
| Limbic lobe | Cingulate gyrus (right) | A24cd caudodorsal area 24 | 4, 6, 38 | 0.70 |
| Frontal lobe | Inferior frontal gyrus (left) | A44v ventral area 44 | -52, 13, 6 | 0.55 |
| Frontal lobe | Inferior frontal gyrus (left) | A44op opercular area 44 | -39, 23, 4 | 0.51 |
| Limbic lobe | Cingulate gyrus (right) | A32p pregenual area 32 | 5, 28, 27 | 0.48 |
| Frontal lobe | Inferior frontal gyrus (right) | A44v ventral area 44 | 54, 14, 11 | 0.47 |
| Insular lobe | Insular gyrus (left) | dIa dorsal agranular insula | -34, 18, 1 | 0.46 |
| Frontal lobe | Precentral gyrus (right) | A4tl area 4 (tongue and larynx region) | 54, 4, 9 | 0.45 |

*Table 7.* Sex-discriminative brain region scores (normalized to [0, 1]) in the visual network (VSN) for HCP-Task (Figure 4 bottom left).

| Lobe | Gyrus (hemisphere) | Region | MNI (x, y, z) | Score |
|---|---|---|---|---|
| Temporal lobe | Parahippocampal gyrus (right) | TH area TH (medial PPHC) | 19, -36, -11 | 0.92 |
| Occipital lobe | Medioventral occipital cortex (left) | vmPOS ventromedial parietooccipital sulcus | -13, -68, 12 | 0.91 |
| Occipital lobe | Medioventral occipital cortex (left) | rCunG rostral cuneus gyrus | -5, -81, 10 | 0.88 |
| Occipital lobe | Medioventral occipital cortex (right) | rLinG rostral lingual gyrus | 18, -60, -7 | 0.58 |
| Occipital lobe | Lateral occipital cortex (right) | OPC occipital polar cortex | 22, -97, 4 | 0.56 |
| Parietal lobe | Inferior parietal lobule (left) | A39c caudal area 39 (PGp) | -34, -80, 29 | 0.55 |
| Limbic lobe | Cingulate gyrus (right) | A23v ventral area 23 | 9, -44, 11 | 0.54 |
| Parietal lobe | Precuneus (right) | dmPOS dorsomedial parietooccipital sulcus (PEr) | 16, -64, 25 | 0.50 |
| Occipital lobe | Medioventral occipital cortex (right) | vmPOS ventromedial parietooccipital sulcus | 15, -63, 12 | 0.40 |
| Temporal lobe | Fusiform gyrus (right) | A37mv medioventral area 37 | 31, -62, -14 | 0.38 |
| Temporal lobe | Parahippocampal gyrus (left) | TL area tl (lateral PPHC, posterior parahippocampa) | -28, -32, -18 | 0.37 |
| Temporal lobe | Fusiform gyrus (right) | A37lv lateroventral area 37 | 43, -49, -19 | 0.37 |
| Occipital lobe | Medioventral occipital cortex (right) | rCunG rostral cuneus gyrus | 7, -76, 11 | 0.36 |
| Occipital lobe | Lateral occipital cortex (right) | iOccG inferior occipital gyrus | 32, -85, -12 | 0.36 |
| Temporal lobe | Parahippocampal gyrus (right) | TL area TL (lateral PPHC, posterior parahippocamp) | 30, -30, -18 | 0.36 |

*Table 8.* Sex-discriminative brain region scores (normalized to [0, 1])in the subcortical network (SCN) for HCP-Task (Figure 4 bottom middle).

| Lobe | Gyrus (hemisphere) | Region | MNI (x, y, z) | Score |
|---|---|---|---|---|
| Subcortical nuclei | Thalamus (right) | mPMtha pre-motor thalamus | 12, -14,  1 | 0.72 |
| Subcortical nuclei | Thalamus (right) | mPFtha medial pre-frontal thalamus | 7, -11,  6 | 0.66 |
| Subcortical nuclei | Thalamus (right) | cTtha caudal temporal thalamus | 10, -14, 14 | 0.65 |
| Insular lobe | Insular gyrus (left) | vIa ventral agranular insula | -32, 14, -13 | 0.61 |
| Subcortical nuclei | Basal ganglia (left) | vCa central caudate | -12, 14,  0 | 0.60 |
| Subcortical nuclei | Basal ganglia (left) | vmPu ventromedial putamen | -23,  7, -4 | 0.46 |
| Subcortical nuclei | Basal ganglia (right) | dlPu dorsolateral putamen | 29, -3,  1 | 0.45 |
| Subcortical nuclei | Thalamus (right) | Otha occipital thalamus | 13, -27,  8 | 0.41 |
| Limbic lobe | cingulate gyrus (left) | A24rv rostroventral area 24 | -3,  8, 25 | 0.36 |

*Table 9.* Sex-discriminative brain region scores (normalized to [0, 1]) in the dorsal attention network (DAN) for HCP-Task (Figure 4 bottom right).

| Lobe | Gyrus (hemisphere) | Region | MNI (x, y, z) | Score |
|---|---|---|---|---|
| Parietal lobe | Superior parietal lobule (right) | A5l lateral area 5 | 35, -42, 54 | 0.92 |
| Frontal lobe | Precentral gyrus (right) | A6cvl caudal ventrolateral area 6 | 51,  7, 30 | 0.51 |
| Parietal lobe | Superior parietal lobule (left) | A7c caudal area 7 | -15, -71, 52 | 0.50 |
| Frontal lobe | Superior frontal gyrus (left) | A6dl dorsolateral area 6 | -18, -1, 65 | 0.42 |
| Temporal lobe | Inferior temporal gyrus (right) | A37elv extreme lateroventral area 37 | 53, -52, -18 | 0.42 |
| Parietal lobe | Superior parietal lobule (right) | A7r rostral area 7 | 19, -57, 65 | 0.41 |
| Parietal lobe | Inferior parietal lobule (right) | A40rd rostrodorsal area 40 (PFt) | 47, -35, 45 | 0.40 |
| Temporal lobe | Middle temporal gyrus (right) | A37dl dorsolateral area 37 | 60, -53,  3 | 0.40 |

