# OpenReview forum: "DynDepNet: Learning Time-Varying Dependency Structures from fMRI Data via Dynamic Graph Structure Learning"
_ICML.cc/2023/Workshop/IMLH — IMLH 2023 Oral_

### Official Review · Reviewer_SyC6 · 2023-06-07
**A novel fMRI classification model that utilizes dynamic functional connectivity and graph neural network**

**Rating:** 9
**Confidence:** 5

**Review:**

### Summary:
This study presents a novel fMRI classification model that utilizes dynamic functional connectivity (dFC) and graph neural network (GNN), arguing that dynamic graph is superior to commonly used static graph. They conducted experiments to classify gender with the large-scale HCP S1200 dataset. Their hyperparameter sensitivity analysis and ablation study are satisfactory, baseline comparison is largely satisfactory but can be improved.

### Strengths:
(1) The study includes satisfactory hyperparameter sensitivity experiments. As time windows length increases to the maximum, the authors' methods learn one static graph. Fig. 2 shows a decreasing trend in performance score when increasing window length and stride.

(2) The authors ablated every proposed module in Table 2, where self attention showed most significant improvement, where temporal attention and regularization is less important.

(3) The authors interpreted sex-discriminative brain region, the interpretability of their methods is a major strength in understanding the underlying black-box classification model, which is crucial in real-world application.

### Weaknesses:
(1) They did comparison to a fruitful of baseline methods demonstrated effectiveness of their method, but the comparison design is less satisfactory, as baseline methods have different inputs (dFC or FC) and number of parameters. For an SVM baseline model, dFC input should have much more information compared to FC input (suppose the BOLD time-series is not passed as input).

### Minor Issues:
(1) Fig. 1 is not self-complete, as readers would have no idea what each notation represents without reading the main text. Adding more description for the notations would help.

(2) Table 1 models could be grouped by static/dynamic/etc. methods.

### Suggestions:
(1) To better understand the effectiveness of using dFC, this study could show how the whole framework performs when time windows are at the maximum (static graph), as demonstrated in Fig. 2 that as time windows increase the performance drops.

(2) Also to better understand the effectiveness of using dFC, the baseline static graph methods (KRR, SVM) can be easily extended to dFC by stacking time windows, or passing the learnt dFC in their methods. Showing their performance on dynamic graph would further convince readers that dFC is superior to static FC.

---

### Official Review · Reviewer_AXqX · 2023-06-16
**Dynamic Graph Structure Learning for fMRI data**

**Rating:** 9
**Confidence:** 4

**Review:**

Summary and Strength:
The paper proposes DynDepNet, an end-to-end trainable model that can learn the optimal time-varying dependency structure from fMRI data in the form of a dynamic brain graph. The authors claim that they are the first to propose and address a dynamic Graph Signal Learning (GSL) problem through GNN-based deep learning on BOLD signals derived from fMRI data. The proposed approach uses spatial-temporal attention to exploit the inherent inter and intra relationships of brain region BOLD signals. The paper reports extensive experiments on two real-world fMRI datasets that demonstrate that DynDepNet achieves state-of-the-art results for biological sex classification.

The paper reports on experiments conducted using real-world resting-state and task fMRI datasets for the task of biological sex classification. The DynDepNet achieves state-of-the-art results and outperforms the best baseline by approximately 8 and 6 percentage points. Additionally, the authors analyze the learned dynamic graphs and highlight prediction-related brain regions that align with existing neuroscience literature.

Weakness:
- To improve reader understanding of the results, it would be beneficial to provide more detailed explanations of the strengths and weaknesses of DynDepNet in comparison to the Pearson correlation, as illustrated in Figure 3 which displays the adjacency matrix from both methods.
- Authors mention that "the true dependency structure between brain regions is known prior to model training". It may be worth considering setting the true dependency structure as the optimization initial point to improve the model training efficiency and model performance.

---

### Official Review · Reviewer_JdzF · 2023-06-17
**Review for Submsson101**

**Rating:** 7
**Confidence:** 4

**Review:**

This paper introduces DynDepNet, a novel end-to-end trainable model for learning optimal time-varying dependency structures from fMRI data. This method addresses the problem of dynamic graph structure learning (GSL) using graph neural network (GNN)-based deep learning on BOLD signals derived from fMRI data. DynDepNet utilizes spatial-temporal attention to exploit the inherent inter and intra relationships of brain region BOLD signals. This attention mechanism allows the model to capture the dynamic nature of brain connectivity and incorporate temporal dependencies, leading to improved performance in biological sex classification tasks.
Pros:
1. Introduces a novel method for learning time-varying dependency structures from fMRI data.
2. Demonstrates robustness across different datasets and generalizability to unseen subjects.

Cons:
1. The paper does not explicitly discuss the computational complexity of the DynDepNet method.
2. The evaluation tasks are restricted on biological sex classification.
3. Lack of comparison with existing GNN-based methods in the field of brain network analysis.

---

### Meta-Review · Program_Chairs · 2023-06-19

**Recommendation:** Accept (Oral)
**Confidence:** 4

**Metareview:**

This is a solid work on fMRI data analysis that propose a novel GNN learning dynamic temporal dependency. All the review comments are positive. The authors can incorporate the constructive feedback into the final version.

---

### Decision · Program_Chairs · 2023-06-20

Accept (Oral)